# A Cross-Sectional Study of SARS-CoV-2 Antibodies and Risk Factors for Seropositivity in Staff in Day Care Facilities and Preschools in Denmark

Kamille Fogh,[a,b,c] Alexandra R. R. Eriksen,[a,b,c] Tine Graakjær Larsen,[g] Rasmus B. Hasselbalch,[a,b,c] Henning Bundgaard,[c,d] Bibi F. S. S. Scharff,[c,e] Susanne D. Nielsen,[c,f] Charlotte S. Jørgensen,[g] Christian Erikstrup,[h,i] Lars Østergaard,[i,j] Svend Ellermann-Eriksen,[i,k] Berit Andersen,[i,l] Henrik Nielsen,[m,n] Isik S. Johansen,[o,p] Lothar Wiese,[q] Lotte Hindhede,[h] Susan Mikkelsen,[h,i] Susanne G. Sækmose,[r] Bitten Aagaard,[s] Dorte K. Holm,[p,t] Lene Harritshøj,[c,e] Lone Simonsen,[u] Thea K. Fischer,[v,w] Fredrik Folke,[a,c,x] Freddy Lippert,[c,x] Sisse R. Ostrowski,[c,e] Thomas Benfield,[c,y] Kåre Mølbak,[g,z] Steen Ethelberg,[g,w] Anders Koch,[c,f,g] Anne-Marie Vangsted,[g] Tyra Grove Krause,[g] Anders Fomsgaard,[g] Henrik Ullum,[g] Robert Skov,[g] Kasper Iversen[a,b,c]

aDepartment of Cardiology, Copenhagen University Hospital, Herlev and Gentofte, Hellerup, Denmark
bDepartment of Emergency Medicine, Copenhagen University Hospital, Herlev and Gentofte, Hellerup, Denmark
cDepartment of Clinical Medicine, University of Copenhagen, Copenhagen, Denmark
dDepartment of Cardiology, Copenhagen University Hospital, Rigshospitalet, Denmark
eDepartment of Clinical Immunology, Copenhagen University Hospital, Rigshospitalet, Denmark
fDepartment of Infectious Diseases, Copenhagen University Hospital, Rigshospitalet, Denmark
gStatens Serum Institut, Copenhagen, Denmark
hDepartment of Clinical Immunology, Aarhus University Hospital, Aarhus, Denmark
iDepartment of Clinical Medicine, Aarhus University, Aarhus, Denmark
jDepartment of Infectious Diseases, Aarhus University Hospital, Aarhus, Denmark
kDepartment of Clinical Microbiology, Aarhus University Hospital, Aarhus, Denmark
lUniversity Research Clinic for Cancer Screening, Randers Regional Hospital, Randers, Denmark
mDepartment of Infectious Diseases, Aalborg University Hospital, Aalborg, Denmark
nDepartment of Clinical Medicine, Aalborg University, Odense, Denmark
oDepartment of Infectious Diseases, Odense University Hospital, Odense, Denmark
pDepartment of Clinical Research, University of Southern Denmark, Odense, Denmark
qDepartment of Infectious Diseases, Zealand University Hospital, Roskilde, Denmark
rDepartment of Clinical Immunology, Zealand University Hospital, Koege, Denmark
sDepartment of Clinical Immunology, Aalborg University Hospital, Aalborg, Denmark
tDepartment of Clinical Immunology, Odense University Hospital, Odense, Denmark
uDepartment of Science and Environment, Roskilde University, Roskilde, Denmark
vDepartment of Clinical Research, North Zealand Hospital, Hillerød, Denmark
wDepartment of Public Health, University of Copenhagen, Copenhagen, Denmark
xCopenhagen Emergency Medical Services, Copenhagen, Denmark
yDepartment of Infectious Diseases, Copenhagen University Hospital, Amager and Hvidovre, Hvidovre, Denmark
zDepartment of Veterinary and Animal Sciences, University of Copenhagen, Frederiksberg, Denmark

Address correspondence to Kamille Fogh, kamille.fogh.01@regionh.dk.

The authors declare no conflict of interest.

**ABSTRACT** The aim of this study was to provide information about immunity against COVID-19 along with risk factors and behavior among employees in day care facilities and preschools (DCS) in Denmark. In collaboration with the Danish Union of Pedagogues, during February and March 2021, 47,810 members were offered a point-of-care rapid SARS-CoV-2 antibody test (POCT) at work and were invited to fill in an electronic questionnaire covering COVID-19 exposure. Seroprevalence data from Danish blood donors (total Ig enzyme-linked immunosorbent assay [ELISA]) were used as a proxy for the Danish population. A total of 21,018 (45%) DCS employees completed the questionnaire and reported their POCT result {median age, 44.3 years (interquartile range [IQR], [32.7 to 53.6]); females, 84.1%}, of which 20,267 (96.4%) were unvaccinated and included in analysis. A total of 1,857 (9.2%) participants tested seropositive, significantly higher than a

seroprevalence at 7.6% (risk ratio [RR], 1.2; 95% confidence interval [CI], 1.14 to 1.27) among 40,541 healthy blood donors (median age, 42 years [IQR, 28 to 53]; males, 51.3%). Exposure at work (RR, 2.9; 95% CI, 2.3 to 3.6) was less of a risk factor than exposure within the household (RR, 12.7; 95% CI, 10.2 to 15.8). Less than 25% of participants reported wearing face protection at work. Most of the participants expressed some degree of fear of contracting COVID-19 both at work and outside work. SARS-CoV-2 seroprevalence was slightly higher in DCS staff than in blood donors, but possible exposure at home was associated with a higher risk than at work. DCS staff expressed fear of contracting COVID-19, though there was limited use of face protection at work.

**IMPORTANCE**   Identifying at-risk groups and evaluating preventive interventions in at-risk groups is imperative for the ongoing pandemic as well as for the control of future epidemics. Although DCS staff have a much higher risk of being infected within their own household than at their workplace, most are fearful of being infected with COVID-19 or bringing COVID-19 to work. This represents an interesting dilemma and an important issue which should be addressed by public health authorities for risk communication and pandemic planning. This study design can be used in a strategy for ongoing surveillance of COVID-19 immunity or other infections in the population. The findings of this study can be used to assess the need for future preventive interventions in DCS, such as the use of personal protective equipment.

**KEYWORDS**   surveillance study, seroprevalence, day care facilities, kindergarten, school, staff, employee, SARS-CoV-2, COVID-19, antibodies, point-of-care test

O n 11 March 2020, COVID-19 was declared a pandemic (1). To prevent the spread of SARS-CoV-2 and ultimately COVID-19, a number of national preventive measures were implemented involving social distancing, increased hygiene, and closure of workplaces (2), including closure of day care facilities (nurseries or kindergartens) and preschools in Denmark (3, 4). This study was done in a period where the society was partially closed, with public gathering restrictions, mandatory use of face masks, and PCR and antigen tests (5). On 27 December 2020, the first COVID-19 vaccines were administered to selected groups. As part of the COVID-19 pandemic management and preventive measures, a complete lockdown of DCS was implemented in Denmark during the first surge (March to May 2020), as children were assumed to play a central role in the spread of SARS-CoV-2 (6). During the second surge (September to December 2020), only schools were closed. From 8 February 2021, preschools were reopened, as children in these age groups were thought to have the least impact on the spread of infection, and returning to school was found crucial for their well-being and academic success (7). Due to educational considerations regarding children's capacity to watch and understand staff members' facial expressions, DCS employees were permitted to use face shields but were not required to use face masks (8).

Evaluation of the effect of these interventions is important for establishment of mitigation strategies in future pandemics. Workplaces are potential sources of SARS-CoV-2 exposure (9). So far, no increased risk of infection in day care facilities and preschools (DCS) has been found compared to the risk in public or private settings, although children exhibit fewer symptoms, which may facilitate virus spread (10). In a recent systematic review of 40 studies, children were found to transmit SARS-CoV-2 infection at a higher rate to adults than to other children, and adults in the household were at the highest risk of transmission from an infected child (11). Household studies from Denmark have further highlighted the household as an important arena of transmission, including transmission from children under 5 years of age to adult household members (12, 13). However, DCS may still pose a risk of transmission from children to adult, and only a few studies, including a small number of cases, have investigated the risk of infection among staff in DCS (14–16).

This study provides information about infection and immunity to COVID-19 among employees working in DCS in Denmark as part of the national large-scale epidemiological surveillance study, Testing Denmark (17). We used a point-of-care rapid antibody test

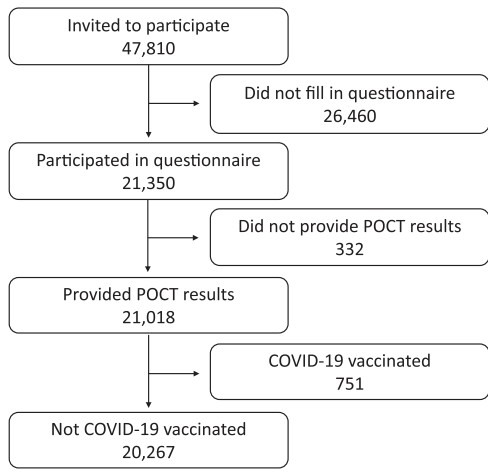

**FIG 1** CONSORT diagram.

(POCT), identifying previous infection by detecting IgM or IgG against the virus spike pro-
tein, allowing an estimate of the seroprevalence along with risk factors and behavior
among DCS staff.

## RESULTS

**Baseline characteristics of the study cohort.** During February and March 2021,
47,810 DCS employees in Denmark were invited to participate. A total of 21,350 (45%)
DCS employees were included and answered the questionnaire, of whom 21,018 (98%)
provided their POCT result (Fig. 1). A total of 751 participants were excluded due to prior
reported COVID-19 vaccination, and 110 participants were excluded due to inconclusive
test results. Baseline characteristics of unvaccinated participants who completed the
questionnaire are presented in Table 1. Included participants had a median age of

**TABLE 1** Baseline characteristics of the study cohort on age, sex, BMI, smoking, alcohol use, education level, and place of work stratified by
seropositivity among 20,267 unvaccinated participants[b]

| Characteristic | Data for patients who were: | | | P value |
| --- | --- | --- | --- | --- |
| | Seronegative | Seropositive | Total | |
| No. of patients | 18,410 | 1,857 | 20,267 | |
| Female (no. [%]) | 15,342 (84.4) | 1,496 (81.4) | 16,838 (84.1) | 0.002 |
| Male (no. [%]) | 2,828 (15.6) | 341 (18.6) | 3,169 (15.8) | |
| Age (median [IQR]) | 44.4 [32.8, 53.7] | 43.9 [31.6, 52.7] | 44.3 [32.7, 53.6] | 0.024 |
| Body mass index (median [IQR]) | 25.3 [22.6, 29.1] | 25.3 [22.8, 29.4] | 25.3 [22.7, 29.1] | 0.155 |
| Ever smoker (no. [%]) | 3,595 (19.5) | 279 (15.0) | 3,874 (19.1) | <0.001 |
| Ever use of alcohol (no. [%])[a] | 15,322 (83.7) | 1,656 (80.6) | 16,978 (83.8) | 0.004 |
| Previously positive PCR test (no. [%]) | 736 (4.3) | 1,066 (61.1) | 1802 (8.9) | <0.001 |
| Education (no. [%]) | | | | 0.005 |
|   No formal education | 191 (1.0) | 15 (0.8) | 206 (1.0) | |
|   Primary education | 663 (3.6) | 63 (3.4) | 726 (3.6) | |
|   Secondary education (youth education) | 1,889 (10.3) | 233 (12.7) | 2,122 (10.5) | |
|   Vocational training or short-/medium-term higher education | 13,316 (72.7) | 1,329 (72.3) | 14,645 (72.7) | |
|   Long-term higher education | 2,078 (11.4) | 191 (10.4) | 2,269 (11.3) | |
| Place of work in Denmark | | | | <0.001 |
|   Capital region of Denmark | 5,152 (28.2) | 713 (38.6) | 5,865 (29.1) | |
|   Region Zealand | 2,178 (11.9) | 248 (13.4) | 2,426 (12.0) | |
|   Southern Denmark | 3,588 (19.6) | 286 (15.5) | 3,874 (19.2) | |
|   Central Denmark | 5,769 (31.5) | 474 (25.7) | 6,243 (31.0) | |
|   Northern region of Denmark | 1,614 (8.8) | 126 (6.8) | 1.740 (8.6) | |

**TABLE 2** Characteristics of the study cohort based on type of employment

| Characteristic | Data for: | | | | | | | |
| | Teachers | Assistants to teacher | Managers | Nonteaching personnel | Substitute/temporary employees | Other | Total | P value |
|---|---|---|---|---|---|---|---|---|
| No. of participants | 10,768 | 4,572 | 1,819 | 357 | 565 | 2,179 | 20,267 | |
| Seropositive (no. [%]) | 960 (8.9) | 455 (10.0) | 162 (8.9) | 42 (11.8) | 55 (9.7) | 182 (8.4) | 1,857 (9.2) | 0.151 |
| IgM (no. [%]) | 366 (3.4) | 175 (3.8) | 61 (3.4) | 14 (3.9) | 20 (3.5) | 61 (2.8) | 697 (3.4) | 0.497 |
| IgG (no. [%]) | 797 (7.4) | 376 (8.2) | 138 (7.6) | 33 (9.2) | 46 (8.1) | 157 (7.2) | 1,548 (7.6) | 0.468 |
| IgM and IgG (no. [%]) | 203 (1.9) | 96 (2.1) | 37 (2.0) | 5 (1.4) | 11 (1.9) | 36 (1.7) | 388 (1.9) | 0.872 |

44.3 years and were more often females (84.1%). Participants were enrolled from all five regions in Denmark (Table 1).

A total of 9.2% of participants tested seropositive, with 7.6% positive for IgG antibodies, 3.4% positive for IgM, as well as 1.9% positive for both IgG and IgM antibodies (Table 2). Seropositive participants had a slightly but significantly younger median age of 43.9 years ($P = 0.02$) and were more likely to be male 18.6% ($P = 0.002$) than seronegative patients. Furthermore, seropositivity was significantly associated with the geographical location of workplace (Table 1), with seropositive participants more likely to work in the eastern part of Denmark (the capital region and region Zealand).

Seroprevalence in the group of Danish blood donors ($n = 40,541$) determined during the same period was 7.6% {median age, 42 years (interquartile range [IQR], 28 to 53); males, 51.3%}. The seroprevalence of 9.2% in DCS was higher than in blood donors (risk ratio [RR], 1.2; 95% confidence interval [CI], 1.14 to 1.27) during the same period.

It should be noted again that different assays were used, as POCT was used in this study, and enzyme-linked immunosorbent assay (ELISA) was used in the blood donors mentioned, so an RR of 1.2 is modest.

**Risk factors.** Figure 2 shows that reporting any kind of exposure to COVID-19 was a risk factor for seropositivity compared to individuals who answered "no" to all categories of exposure in the questionnaire. Being exposed to COVID-19 by a household member was associated with the highest risk (RR, 12.7; 95% CI, 10.2 to 15.8) compared to nonexposed individuals. Physical contact with a person with COVID-19 was also associated with a high risk of seropositivity (RR, 6.5; 95% CI, 5.2 to 8.1). as was 15 min of close contact with a person positive for SARS-CoV-2 (RR, 5.0; 95% CI, 4.0 to 6.2). Exposure through work and exposure to a family member was associated with a similar level of risk (RR, 2.9; 95% CI, 2.3 to 3.6, and RR, 3.1; 95% CI, 2.5 to 3.9, respectively). The size of households varied from one to six people, but no observed significant association between household size and seropositivity was found ($P = 0.74$). Finally, seropositive DCS staff members were less likely to consume alcohol ($P = 0.004$) and smoke ($P < 0.001$) than seronegative members (Table 1).

**Type of employment and working place.** No significant difference in serostatus was found when stratifying by job category ($P = 0.12$) or type of workplace ($P = 0.05$) (Tables 2 and 3). The job category with the largest proportion of seropositive individuals was staff who were not directly involved with taking care of children (11.8%), and the seroprevalence (9.8%) was highest for participants working in day care nurseries for infants or small children, but none of these findings were statistically significant.

**Use of personal protective equipment against COVID-19 at work.** Most DCS staff reported use of protective measures at work (Fig. 3), especially frequent handwashing (96%, $n = 19,449$), though less than 25% reported wearing a face mask or face shield at work. No significant difference in serostatus was observed between individuals specifying the use of the different individual protective measures. A significant association was, however, found between serostatus and answering "yes" to not using any of the protective measures inquired upon against COVID-19 at work ($P = 0.001$). This was reported by 2.8% of the seropositive and 1.7% of seronegative DCS employees.

**Fear of COVID-19 infection.** Contracting COVID-19 was a concern among most of the study cohort (Fig. 4). A total of 91.4% ($n = 17,439$) of DCS staff feared contracting

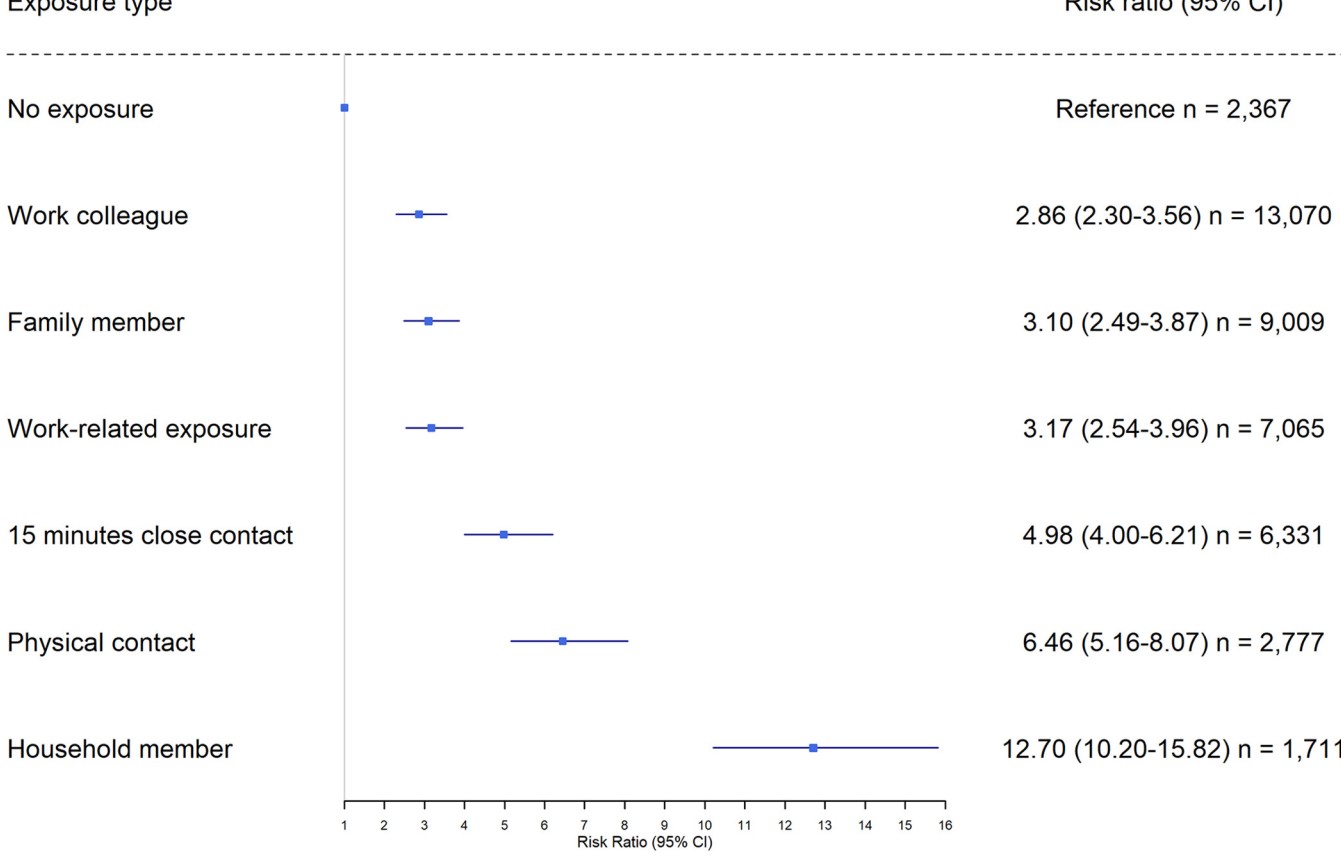

**FIG 2** Risk factors stratified by seropositivity among 20,267 unvaccinated participants.

infection at work, 90.0% (*n* = 17,173) of staff feared contracting infection outside work, 92.5% (*n* = 17,640) feared transmission of infection from workplace to household, and 88.5% (*n* = 16,892) feared transmission of infection from household to workplace. The proportion of worried individuals in the seronegative group was significantly larger than the proportion within the seropositive group, as shown in Table 4 and Fig. 4. For example, 15,956 (92.0%) of the seronegative individuals and 1,483 (85.9%) of the seropositive individuals were worried about contracting COVID-19 at work (*P* < 0.001).

## DISCUSSION

In this nationwide, cross-sectional study of 21,018 DCS staff members, we found that the seroprevalence among DCS staff members (9.2%) was significantly higher than among blood donors (7.6%) in the same period. Less than 25% of staff members wore

**TABLE 3** Frequencies of seropositivity stratified according to workplace

| Characteristic | Data for patients at workplace: | | | | | | | | |
|---|---|---|---|---|---|---|---|---|---|
| | Day nursery for infants/small children | Kindergarten | Combined daycare center | Before and after school care | School | Special within public daycare facility | Special within public school facility | Other | P value |
| No. of participants | 2,881 | 4,503 | 5,549 | 2,121 | 2,066 | 524 | 1,042 | 987 | |
| Seropositive (no. [%]) | 281 (9.8) | 373 (8.3) | 522 (9.4) | 192 (9.1) | 186 (9.0) | 47 (9.0) | 87 (8.3) | 94 (9.5) | 0.051 |
| IgM (no. [%]) | 108 (3.7) | 144 (3.2) | 204 (3.7) | 65 (3.1) | 64 (3.1) | 22 (4.2) | 33 (3.2) | 34 (3.4) | 0.685 |
| IgG (no. [%]) | 240 (8.3) | 306 (6.8) | 433 (7.8) | 166 (7.8) | 159 (7.7) | 37 (7.1) | 72 (6.9) | 75 (7.6) | 0.113 |
| IgM and IgG (no. [%]) | 67 (2.3) | 77 (1.7) | 115 (2.1) | 39 (1.8) | 37 (1.8) | 12 (2.3) | 18 (1.7) | 15 (1.5) | 0.537 |

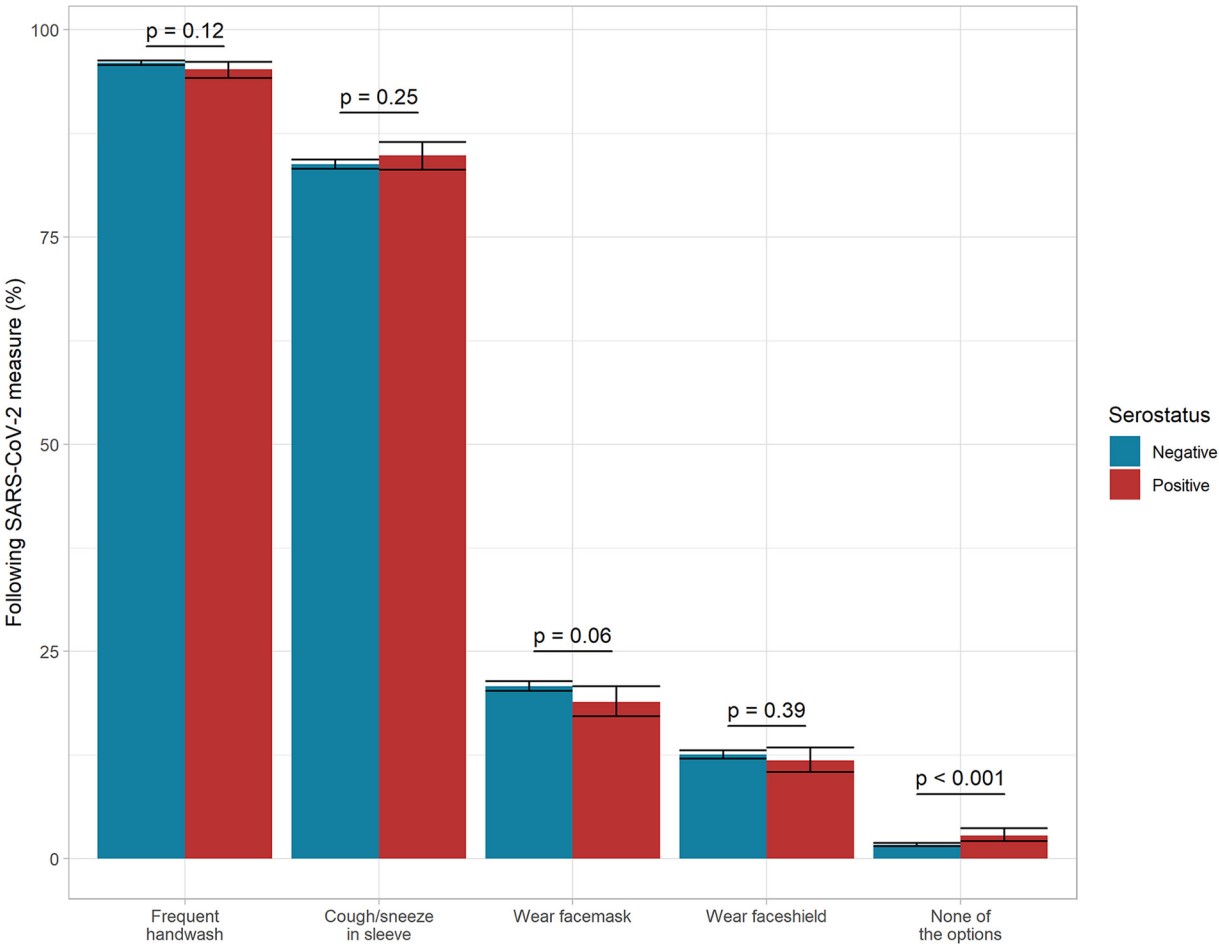

**FIG 3** Proportions of participants following public health measures and wearing personal protective equipment at work stratified for serostatus among 20,267 unvaccinated participants.

face masks or face shields at work. Exposure at work was less of a risk factor than exposure within the household. A fear of contracting COVID-19 was frequently reported by DCS staff members, especially among seronegative participants.

In comparison to the above-mentioned seroprevalences, the March 2021 results of the Danish national seroprevalence study, where 10,631 randomly selected Danish residents above 12 years of age had antibodies measured by use of the Wantai ELISA, showed a seroprevalence of 7.0% (95% CI, 6.6 to 7.4%) (18).

**Workplace and exposure.** This study showed an elevated risk among DCS staff, which is contrary to observations from other studies, where no elevated risk of contracting or spreading COVID-19 was found for DCS staff (14, 16, 19). A study from Germany (318 children, 299 parents, and 233 childcare employees) found DCS staff at higher risk of infection than children, suggesting that transmission of infection is more common between adults than between children and adults and that DCS settings are not crucial in driving the SARS-CoV-2 pandemic (20). Individuals working in DCS will have different job functions that entail different levels of contact with the children, and it would therefore be expected that serostatus within job categories would reflect this. In our study, no significant difference was found between job categories or type of workplace and seropositivity, which may suggest that the spread of infection is not being driven by the children but by some other source.

**Behavior and risk factors.** Despite a high level of adherence to national recommendations (see Fig. S1 in the supplemental material), no discernible difference was found between individual protective health measures and serostatus, as seen in earlier studies (17, 21), but no adherence to all of the measures was significantly associated with seropositivity. Several studies have reported low usage of face masks in DCS. A

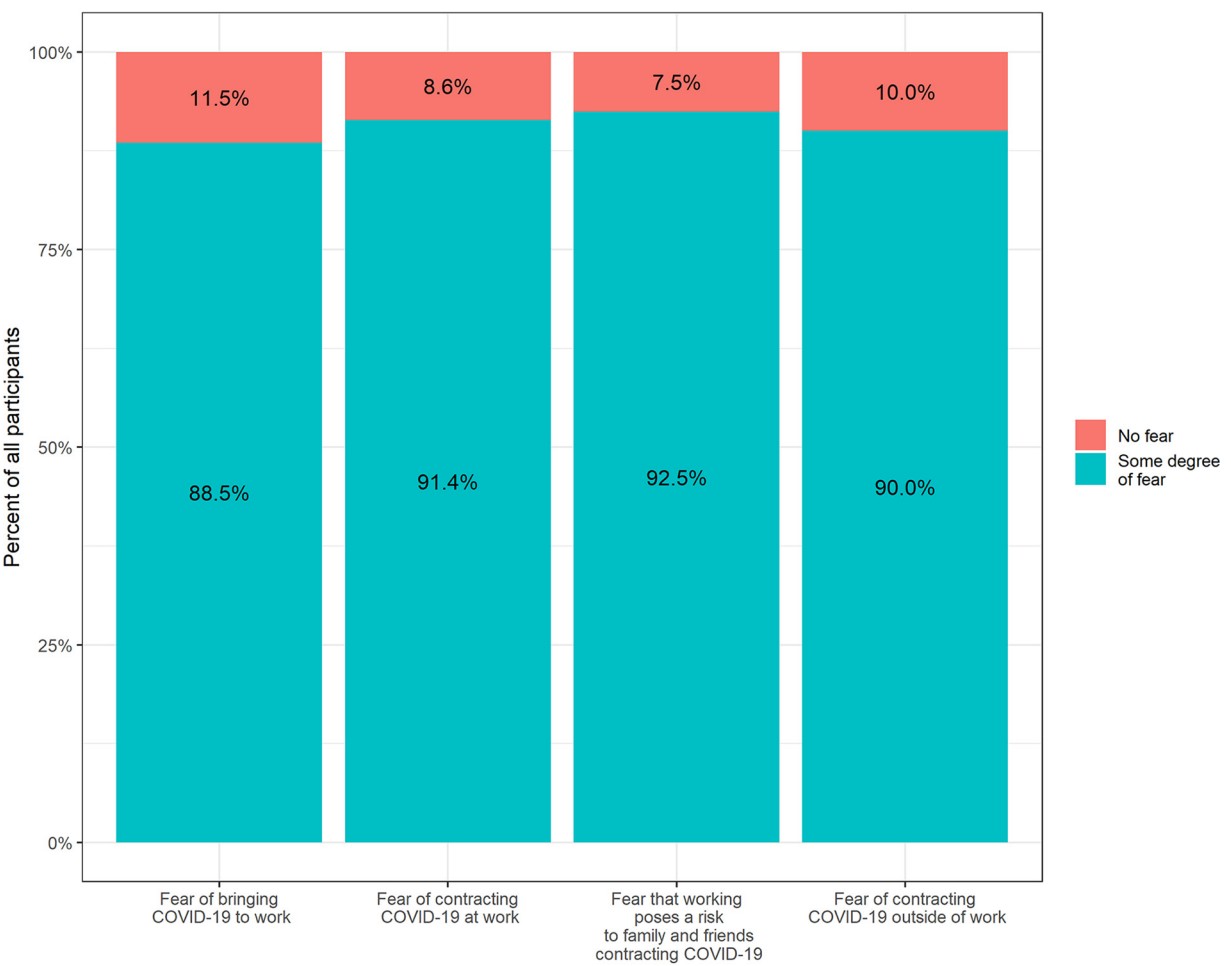

**FIG 4** Fear of COVID-19 among 19,077 unvaccinated participants. A total of 1,190 did not answer questions concerning fear in the questionnaire.

U.S.-based study showed that only 35% of childcare providers reported using face masks at work (14), while a British study demonstrated a low predicted probability of 51% of education workers wearing face coverings during close contact (9). In our study, less than 25% of participants wore face masks at work, while even fewer participants wore face shields, with a tendency that more seronegative than seropositive participants wore face masks. As previously stated, DCS employees were not required to wear face masks at work, but they could use face shields if they wished. Small children, in particular, need to see and read facial expressions or body language when in contact with adults.

**TABLE 4** Frequencies of fear of COVID-19 among 19,077 unvaccinated participants

| Characteristic | Level[a] | Seronegative (no. [%]) | Seropositive (no. [%]) | Total (no. [%]) | P |
|---|---|---|---|---|---|
| No. | | 17,350 | 1,727 | 19,077 | |
| Fear of bringing COVID-19 to work | 2 | 15,404 (88.8) | 1,488 (86.2) | 16,892 (88.5) | 0.001 |
| | 1 | 1,946 (11.2) | 239 (13.8) | 2,185 (11.5) | |
| Fear of contracting COVID-19 at work | 2 | 15,956 (92.0) | 1,483 (85.9) | 17,439 (91.4) | <0.001 |
| | 1 | 1,394 (8.0) | 244 (14.1) | 1,638 (8.6) | |
| Fear that working poses a risk to family and friends contracting COVID-19 | 2 | 16,077 (92.7) | 1,563 (90.5) | 17,640 (92.5) | 0.001 |
| | 1 | 1,273 (7.3) | 164 (9.5) | 1,437 (7.5) | |
| Fear of contracting COVID-19 outside work | 2 | 15,706 (90.5) | 1,467 (84.9) | 17,173 (90.0) | <0.001 |
| | 1 | 1,644 (9.5) | 260 (15.1) | 1,904 (10.0) | |

[a]Level 1, no fear; level 2, some degree of fear.

Face masks were recommended for use in public transportation and grocery stores at the time of the study. People who use face masks more frequently in society may also protect themselves more frequently at work and thus are not only seronegative due to the use of face masks at work but also in general, which reduces the risk of becoming infected.

All risk factors (exposure types) surveyed were found to be associated with increased risk of infection. Living in a household with a SARS-CoV-2-infected person increased the risk of infection substantially and more so than exposure at work. Several studies have found that transmission within households plays an important role in SARS-CoV-2 transmission (17, 22, 23), which is why the household is an area with potential for prevention. Alcohol consummation and smoking were less likely among seropositive people. The same pattern was observed in a survey conducted in social housing areas as part of the Testing Denmark study in January 2021 (22). Seropositivity increased with age in the social housing survey, and the proposed reason for the lower likelihood to smoke or consume alcohol if seropositive was related to the fact that these risk factors are expected to be more prevalent among younger people. In our study, we found that seropositive participants were younger in age, which means that age cannot be the reason. A possible explanation for the lower likelihood of smoking or drinking alcohol if seropositive could be that smokers and alcohol users were less likely to participate in the study.

**Fear of COVID-19 at the workplace.** The fear among health care workers (HCWs) during the COVID-19 pandemic has been described (24) and is innate to health care. However, fear among DCS staff due to workloads or sudden changes in routine following mitigation strategies has not been described during this pandemic. According to the Canadian EnCORE study, 65% of DCS staff were concerned about contracting COVID-19 at work (25), and a survey in the United States found 77% of employees were concerned about contracting COVID-19 at work (14). In comparison, this study found a much higher percentage of concerned individuals among DCS staff, with 91.4% of them concerned about contracting COVID-19 at work and 92.5% concerned that work could pose a risk of spreading SARS-CoV-2 to family members and friends. Frontline HCWs caring for COVID-19 patients are reported to be less fearful about becoming infected than HCWs working in other units. This perplexing finding could be linked to a lack of education or communication (24). Concerns about contracting or spreading COVID-19 among DCS staff at work, even though we found the highest risk of infection when living in a household with an infected individual, may be related to suboptimal communication and education from the authorities providing the mitigation strategies, which could be a focal point in future strategies.

**Strength and limitations.** This large-scale study had a high participation rate (45%) and widespread national participation with the possibility of including both low- and high-seroprevalence DCS during the study period.

The possibility of misclassification bias as participants read the POCT themselves is one of the study's limitations, which could result in false-negative or false-positive test results. Participants with or without prior SARS-CoV-2 infection may be more or less likely to participate, potentially resulting in sampling bias. Furthermore, seroprevalence in blood donors is determined using an ELISA, whereas seroprevalence in this study was determined using POCT results. Seropositivity may be lower than expected as a result of the healthy worker effect because blood donors are typically healthier than the general population. The time delay could potentially be a source of bias since participants who were recently infected may not get a positive POCT result due to a lack of antibodies. Finally, DCS that are not affiliated with the Danish Union of Pedagogues (BUPL) are deselected, implying that selection bias cannot be ruled out, but as stated in Materials and Methods, BUPL represents nearly all DCS staff members.

This study design is novel and can be used as a supplemental model in a future general test strategy for ongoing surveillance of COVID-19 immunity or other infections in the population. The findings of this study can be used to assess the need for

future preventive interventions in DCS, such as the use of personal protective equipment.

**Conclusion.** SARS-CoV-2 seroprevalence was slightly higher in day care and preschool staff than in blood donors in Denmark, but possible exposure at home was associated with a higher risk than at the place of work. DCS staff, particularly those who were seronegative, largely expressed fear of contracting COVID-19, though there was limited use of face masks or face shields at work. The contrast between perceived fear (being infected at work) and measured risk (being infected in the household) represents an important issue for risk communication as well as pandemic planning.

## MATERIALS AND METHODS

**Study design and participation.** In collaboration with the Danish Union of Pedagogues (BUPL), a nationwide cross-sectional surveillance study among 47,810 union members and employees over the age of 15 years at DCS was carried out in February and March 2021 as part of the nationwide large-scale study Testing Denmark (17, 22, 26). Participants from selected DCS were invited to perform an individual POCT at work, identifying previous infection by detecting IgM and/or IgG antibodies and responding to an electronic questionnaire in the Research Electronic Data Capture database (REDCap). Employees affiliated with BUPL represent both municipal, self-governing, small, or large DCS and work with children aged 0 to 10 years old. In Denmark, 60,136 individuals are working in DCS (27), and approximately 60.000 are members of the BUPL (28).

Invitation letters and test material (POCT, a small container of detection buffer, capillary tubes, and finger prickers) were sent to DCS members in Denmark.

The POCT (CTK Biotech inc., Poway, CA, USA) was performed as a self-test and read by participants individually. POCT sensitivity and specificity were reported to be 96.9% (95% CI, 96.7 to 98.5%) and 99.4% (95% CI, 97.8 to 99.8%) by the manufacturer, respectively (29). A comparative study (cases, 30 people; controls, 10) revealed a slightly lower sensitivity of 90.0% and a 100% specificity (30). A study of 129 nonhospitalized versus 31 hospitalized SARS-CoV-2 patients revealed that the Wantai assay had a sensitivity of 96.7 (95% CI, 92.4 to 98.6) and a specificity of 99.5 (95% CI, 98.7 to 99.8) (31).

A call center was established for participants to contact if they had any questions. The project homepage (www.vitesterdanmark.dk) included information about the project as well as detailed information about the test procedure in Danish.

The questionnaire included questions about risk factors, symptoms, household members, employment, and behavior according to recommendations from the Danish Health Authority (see Fig. S1 in the supplemental material).

As a proxy for the general Danish population, we obtained access to data on seroprevalence in the same period from a convenience sample of unvaccinated Danish blood donors (SARS-CoV-2 total Ig ELISA; Wantai Biological Pharmacy Enterprise, Co, Ltd.), with a median age of 42 years, and 51.3% were male (31, 32).

**Outcome measures.** The outcome was to provide an estimate of the seroprevalence of antibodies against SARS-CoV-2 as a proxy for previous infection among staff in DCS.

**Approvals, ethics, and registrations.** Under the authority task of the Danish national infectious disease control institute, Statens Serum Institut (SSI), the study was carried out as a national surveillance study, which does not require approval from an ethics committee according to Danish law. The study followed the Helsinki II declaration and was registered with the Danish Data Protection Authorities (P-2020-901), which declared that all personal data obtained in REDCap were kept in compliance with the general data protection regulation and data protection law. The information was self-reported, and participation was voluntary. The invitation letter included information on the invitees' legal rights as well as information about the intended usage of their data.

**Statistical analyses.** Participants were categorized as seropositive if they reported being positive for IgG and/or IgM antibodies on POCT. Inconclusive test results (no control line appeared, or the reading chamber was discolored by blood) were excluded, as participants had the possibility of repeating the test. A small number of participants specified that they were vaccinated against SARS-CoV-2, which would give them antibodies against the spike protein and results in a positive POCT result without them having been infected. Individuals stating that they had received SARS-CoV-2 vaccination were excluded from analysis. Individuals were also excluded from analysis of questionnaire questions that they had not answered.

Baseline characteristics of the cohort are presented as numbers (percent) for factors and medians (IQRs) for numeric variables, as appropriate. The Wilcoxon rank-sum test and chi-square test were used for comparisons of groups for continuous and categorical values, respectively.

Risk factors of seropositivity were explored by calculation of crude risk ratio (RR) with 95% confidence intervals (95% CI) and tested for significant differences using Fisher's exact test. When assessing the risk of seropositive POCT results associated with different COVID-19 exposure categories, the risk within each individual exposure category was compared to the risk within the group of individuals specifying no exposure from any of the categories. Body mass index (BMI) was calculated from self-reported height and weight. Self-reported zip codes for working place were grouped according to the five Danish regions. Participants who self-registered as either "day care assistant" or "helper" were grouped into "teacher's assistant." The questionnaire answers, "do not know" and "not relevant," were classified as

"NA." *P* values of <0.05 were considered statistically significant. Data management, statistical analyses, and figures were performed and created using R version 3.6.1 (www.r-project.org).

**Data availability.** The data sets used and analyzed during the current study are available from corresponding author on reasonable request. The data are not publicly available due to Danish regulation.

## SUPPLEMENTAL MATERIAL

Supplemental material is available online only.

**SUPPLEMENTAL FILE 1**, PDF file, 0.1 MB.

## ACKNOWLEDGMENTS

We thank everyone who participated in this study. We also thank BUPL for their assistance in sending out test kit material.

We declare no potential conflict of interest with respect to the research, authorship, and/or publication of this article.

This study was supported by Trygfonden (152144) and Helsefonden (20-A-0124) grants. The funders did not influence study design, conducting, or reporting. The Union of Pedagogues (BUPL) grants covered POCT shipping as well as the call center. BUPL did influence the study design but not conducting or reporting.

The study was designed and initiated by K.F. and K.I. Statistical analysis and visualization were done by: K.F., A.R.R.E., and T.G.L. K.F., A.R.R.E., T.G.L., R.B.H., and K.I. had full access to the data. The first draft was written by K.F., A.R.R.E., T.G.L., H.B., and K.I. All authors have critically revised the manuscript. All authors took part in conceptualization, interpretation, and discussion of results; agree to be accountable for all aspects of the work; and approved the final version of the manuscript.

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
