## [Reviewer comments · Microbiology Spectrum]

Microbiology Spectrum

A cross-sectional study of SARS-CoV-2 antibodies and risk factors for seropositivity in staff in day-care facilities and pre-schools in Denmark

Kamille Fogh, Alexandra Eriksen, Tine Larsen, Rasmus Hasselbalch, Henning Bundgaard, Bibi Scharff, Susanne Nielsen, Charlotte Jørgensen, Christian Erikstrup, Lars Østergaard, Svend Ellermann-Eriksen, Berit Andersen, Henrik Nielsen, Isik Johansen, Lothar Wiese, Lone Simonsen, Lotte Hindhede, Susan Mikkelsen, Thea Fischer, Susanne Sækmose, Bitten Aagaard, Dorte Holm, Lene Harritshøj, Fredrik Folke, Freddy Lippert, Sisse Ostrowski, Thomas Benfield, Kåre Mølbak, Steen Ethelberg, Anders Koch, Anne-Marie Vangsted, Tyra Krause, Anders Fomsgaard, Henrik Ullum, Robert Skov, and Kasper Iversen

Corresponding Author(s): Kamille Fogh, Herlev and Gentofte Hospital, University of Copenhagen

Review Timeline:

Submission Date:	October 13, 2022
Editorial Decision:	November 4, 2022
Revision Received:	December 5, 2022
Accepted:	December 6, 2022

Editor: Holly Ramage

Reviewer(s): Disclosure of reviewer identity is with reference to reviewer comments included in decision letter(s). The following individuals involved in review of your submission have agreed to reveal their identity: James N Moy (Reviewer #1); Jonathan Daniel Hulse (Reviewer #2); Kenji Ota (Reviewer #3)

Transaction Report:

DOI: <https://doi.org/10.1128/spectrum.04174-22>

November 4, 2022

Dr. Kamille Fogh
Herlev and Gentofte Hospital, University of Copenhagen
Dept. of Cardiology & Dept. of Emergency Medicine
Herlev
Denmark

Re: Spectrum04174-22 (A cross-sectional study of SARS-CoV-2 antibodies and risk factors for seropositivity in staff in day-care facilities and pre-schools in Denmark)

Dear Dr. Kamille Fogh:

Thank you for submitting your manuscript to Microbiology Spectrum. Your manuscript was evaluated by three reviewers and the consensus was generally positive. The reviewers felt that the study is of high importance and that your conclusions are largely supported by the data. However, there are some points raised during review that need to be addressed, including some data analyses, as well as further discussion of the significance of the work. When submitting the revised version of your paper, please provide (1) point-by-point responses to the issues raised by the reviewers as file type "Response to Reviewers," not in your cover letter, and (2) a PDF file that indicates the changes from the original submission (by highlighting or underlining the changes) as file type "Marked Up Manuscript - For Review Only". Please use this link to submit your revised manuscript - we strongly recommend that you submit your paper within the next 60 days or reach out to me. Detailed instructions on submitting your revised paper are below.

Link Not Available

Sincerely,

Holly Ramage

Journals Department
Reviewer comments:

Reviewer #1 (Comments for the Author):

Fogh, et al conducted a study with a very large cohort study of day care and preschool workers to determine seroconversion to SARS-CoV-2 and to provide information about risk factors and behavior that might result in COVID-19 infection. 47,810 union members of day care facilities and preschools (DCS) in Denmark were offered a point-of care rapid SARS-CoV-2 antibody test (POCT) at work and were asked to fill out a questionnaire regarding risks of COVID-19 exposure and use of protective

measures. Seroprevalence data from Danish blood donors were used as a proxy for the Danish population. 21,018 (45%) DCS employees completed the questionnaire and reported their POCT results.

This is a very important study that was well designed. The data generated from this study reinforces the fact that the risk of exposure (and infection) is lower in childcare and school settings than in the household. These findings would be useful in the guiding policies regarding keeping schools and childcare facilities open during future COVID-19 surges.

Comments:

In line 155 and 156, the authors state "Seropositive participants were significantly 156 younger median age 43.9 ($p=0.02$)" Although the ages between the groups are statistically significant, I don't think that mean ages of 43.9 and 44.4 are different in terms of human physiology.

Looking at Table 1, it appears that smokers and alcohol users were less likely to be seropositive. I would have expected them to be more likely to be seropositive. Would the authors like to discuss their findings regarding smoking and alcohol use and seropositivity?

Reviewer #2 (Comments for the Author):

Line 28: Replace 'and' with 'as well as'

Line 52-53: Try to avoid using 'and' multiple times in a sentence

Line 54: Try to avoid using 'and' multiple times in a sentence

Line 60 - 61: Try to avoid using 'and' multiple times in a sentence

Line 66 - 67: Run on sentence, Try to avoid using 'and' multiple times in a sentence

Line 78 - 79: Try to avoid using 'and' multiple times in a sentence

Line 83: Use a comma after (BUPL)... Ex: (BUPL),

Line 89: Try to avoid using 'and' multiple times in a sentence

Line 97: Try to avoid using 'and' multiple times in a sentence

Line 155: Try to avoid using 'and' multiple times in a sentence

Line 227: Use a period after wished. Start a new sentence.

Line 228: Try to avoid using 'and' multiple times in a sentence

Line 236-237: Try to avoid using 'and' multiple times in a sentence, Run on sentence.

Line 247-248: Try to avoid using 'and' multiple times in a sentence

Line 254: use a comma instead of a ;

Line 254: After the comma (see above), substitute 'which' instead of 'this'

Line 272-274: Try to avoid using 'and' multiple times in a sentence

Line 297 - 298: Try to avoid using 'and' multiple times in a sentence

Reviewer #3 (Comments for the Author):

1. Though not statistically significant ($p=0.06$), there is a tendency that more participants wore facemasks in seronegative group than seropositive (Figure 3). I would like to know how the authors would consider this tendency and usefulness of wearing masks.

2. Relevant to comment #1, I would like to see the Figure comparing seropositive/negative ratio among those with/without following each preventive intervention.
3. Methods; As in Reference #18&22, sensitivity and specificity of both assays should be described. It will help readers interpreting the results.
4. Table 1; the calculation of the proportion is wrong (for example; Previously positive PCR test among Seropositive = 1066/1857 = 57.4% but 61.1% in Table 1). The table needs revision.

Staff Comments:

Preparing Revision Guidelines

Please return the manuscript within 60 days; if you cannot complete the modification within this time period, please contact me. If you do not wish to modify the manuscript and prefer to submit it to another journal, please notify me of your decision immediately so that the manuscript may be formally withdrawn from consideration by Microbiology Spectrum.

Fogh, et al conducted a study with a very large cohort study of day care and preschool workers to determine seroconversion to SARS-CoV-2 and to provide information about risk factors and behavior that might result in COVID-19 infection. 47,810 union members of day care facilities and preschools (DCS) in Denmark were offered a point-of care rapid SARS-CoV-2 antibody test (POCT) at work and were asked to fill out a questionnaire regarding risks of COVID-19 exposure and use of protective measures. Seroprevalence data from Danish blood donors were used as a proxy for the Danish population. 21,018 (45%) DCS employees completed the questionnaire and reported their POCT results.

This is a very important study that was well designed. The data generated from this study reinforces the fact that the risk of exposure (and infection) is lower in childcare and school settings than in the household. These findings would be useful in the guiding policies regarding keeping schools and childcare facilities open during future COVID-19 surges.

Comments:

In line 155 and 156, the authors state “Seropositive participants were significantly 156 younger median age 43.9 (p=0.02)”

Although the ages between the groups are statistically significant, I don't think that mean ages of 43.9 and 44.4 are different in terms of human physiology.

Looking at Table 1, it appears that smokers and alcohol users were less likely to be seropositive. I would have expected them to be more likely to be seropositive. Would the authors like to discuss their findings regarding smoking and alcohol use and seropositivity?

December 1, 2022

To Editor

Holly Ramage

Microbiology Spectrum

Dear Holly Ramage

Thank you for the opportunity to revise and improve our manuscript

“A cross-sectional study of SARS-CoV-2 antibodies and risk factors for seropositivity in staff in day-care facilities and pre-schools in Denmark”.

We appreciate the time and effort that the reviewers have spent on the revision of our manuscript, and we have carefully addressed all comments and concerns from the reviewers in the point-by-point reply below.

We consider the paper much improved and hope that you will consider it for publication in **Microbiology Spectrum**.

Sincerely yours

Kamille Fogh, MD, Ph.d. student

Department of Cardiology and Department of Emergency Medicine

Herlev-Gentofte Hospital

Borgmester Ib Juuls Vej 1

DK - 2730 Herlev

T +45 2679 8310

E kamille.fogh.01@regionh.dk

Placement of revision (highlighted in red) in each response refers to placement in clean version.

Reviewer #1:

In line 155 and 156, the authors state "Seropositive participants were significantly younger median age 43.9 (p=0.02)"

Although the ages between the groups are statistically significant, I don't think that mean ages of 43.9 and 44.4 are different in terms of human physiology.

Looking at Table 1, it appears that smokers and alcohol user were less likely to be seropositive. I would have expected them to be more likely to be seropositive. Would the authors like to discuss their findings regarding smoking and alcohol use and seropositivity?

Response to Reviewer:

Thank you for this comment.

We can agree that median ages of 43.9 and 44.4 are not that different and have added a sentence to the manuscript.

A survey in social housing areas was conducted as part of the Testing Denmark study in January 2021, and the same trend was observed, with seropositive people being less likely to consume alcohol (p=0.04) and smoke (p0.001). Seropositivity increased with age in the social housing study, and a proposed reason that smokers and alcohol users were less likely to be seropositive was that alcohol consumption and smoking were expected to be more prevalent among young people, who had a lower risk of seropositivity. Another explanation given in that study was that people with such risk factors did not participate. Because our study found that seropositive participants had a significantly lower median age of 43.9 (p=0.02), the age explanation cannot be used to answer the question above. However, one possible explanation is that smokers and alcohol users were less likely to participate.

Ref: Fogh K, Eriksen ARR, Hasselbalch RB, Kristensen ES, Bundgaard H, Nielsen SD, Jørgensen CS, Scharff BFSS, Erikstrup C, Sækmose SG, Holm DK, Aagaard B, Norsk J, Nielsen PB, Kristensen JH, Østergaard L, Ellermann-Eriksen S, Andersen B, Nielsen H, Johansen IS, Wiese L, Simonsen L, Fischer TK, Folke F, Lippert F, Ostrowski SR, Ethelberg S, Koch A, Vangsted AM, Krause TG, Fomsgaard A, Nielsen C, Ullum H, Skov R, Iversen K. Seroprevalence of SARS-CoV-2 antibodies in social housing areas in Denmark. BMC Infect Dis. 2022 Feb 10;22(1):143. doi: 10.1186/s12879-022-07102-1. PMID: 35144550; PMCID: PMC8830972.

Prior to revision:

Participants were significantly younger median age 43.9 (p=0.02) and more likely to be male 18.6 % (p=0.002) than seronegative.

Revision (Results, page 10, l. 159):

Seropositive participants were slightly but significantly younger median age 43.9 (p=0.02) and more likely to be male 18.6 % (p=0.002) than seronegative.

Added to the manuscript (Discussion, page 14, l. 243):

Alcohol consumption and smoking were less likely among seropositive. The same pattern was observed in a survey conducted in social housing areas as part of the Testing Denmark study in January 2021 (ref).

Seropositivity increased with age in the social housing survey, and the proposed reason for the lower likelihood to smoke or consume alcohol if seropositive was related to the fact that these risk factors are expected to be more prevalent among younger people. In our study, we found that seropositive participants were younger in age, which means that age cannot be the reason. A possible explanation for the lower likelihood of smoking or drinking alcohol if seropositive could be that smokers and alcohol users were less likely to participate in the study.

Ref: Fogh K, Eriksen ARR, Hasselbalch RB, Kristensen ES, Bundgaard H, Nielsen SD, et al. Seroprevalence of SARS-CoV-2 antibodies in social housing areas in Denmark. BMC Infect Dis. 2022;22(1):143.

Reviewer #2:

Thank you for these very relevant sentence-change suggestions. We have corrected the manuscript in accordance with these, see changes in red below.

Line 52-53: Try to avoid using 'and' multiple times in a sentence

To prevent the spread of SARS-CoV-2 and ultimately COVID-19, a number of national preventive measures were implemented involving social distancing, increased hygiene, closure of workplaces(1) including closure of day-care facilities (nurseries ~~and-or~~ kindergartens) and pre-schools in Denmark

Line 54: Try to avoid using 'and' multiple times in a sentence

This study was done in a period where the society was partially closed with public gathering restrictions, mandatory use of face masks ~~and-use-of~~, PCR and antigen tests (2).

Line 60 - 61: Try to avoid using 'and' multiple times in a sentence

From February 8, 2021 pre-schools were reopened, as children in these age groups were thought to have the least impact on the spread of infection, ~~as and~~ returning to school was found crucial for their wellbeing and academic success (3).

Line 66 - 67: Run on sentence, Try to avoid using 'and' multiple times in a sentence

So far, no increased risk of infection in day-care facilities and pre-schools (DCS) have been found compared to the risk in public or private settings, although children exhibit fewer symptoms, ~~which and thereby~~ may facilitate virus spread (4).

Line 78 - 79: Try to avoid using 'and' multiple times in a sentence

We used point-of-care rapid antibody test (POCT), identifying previous infection by detecting immunoglobulin (Ig) M ~~or and~~ IgG against the virus spike protein allowing an estimate of the seroprevalence along with risk factors and behavior among DCS staff.

Line 83: Use a comma after (BUPL)... Ex: (BUPL),

In collaboration with the Danish Union of Pedagogues (BUPL), a nationwide, cross-sectional, surveillance study among 47,810 union members and employees over the age of 15 years at DCS was carried out in February and March 2021, as part of the nationwide large-scale study "Testing Denmark" (5-7).

Line 89: Try to avoid using 'and' multiple times in a sentence

Employees affiliated to BUPL represent both municipal, self-governing, small, ~~or and~~ large DCS and work with children aged 0-10 years old.

Line 97: Try to avoid using 'and' multiple times in a sentence

The project homepage (www.vitesterdanmark.dk) included ~~information~~ about the project ~~as well as and~~ detailed information ~~about the on-the~~ test-procedure ~~was available~~ in Danish ~~and provided on the project homepage (www.vitesterdanmark.dk).~~

Line 155: Try to avoid using 'and' multiple times in a sentence

A total of 9.2% participants tested seropositive, with 7.6% positive for IgG antibodies, 3.4% positive for IgM ~~as well as and~~ 1.9% positive for both IgG and IgM antibodies (table 2).

Line 227: Use a period after wished. Start a new sentence.

As previously stated, DCS employees were not required to wear facemasks at work, but they could use a face shield if they wished. ~~As s~~Small children, in particular, will need to see and read facial expressions and body language when in contact with adults.

Line 228: Try to avoid using 'and' multiple times in a sentence

As previously stated, DCS employers were not required to wear facemasks at work, but they could use a face shield if they wished. Small children, in particular, will need to see and read facial expressions ~~or and~~ body language when in contact with adults.

Line 236-237: Try to avoid using 'and' multiple times in a sentence, Run on sentence.

The fear among health care workers (HCW) during the COVID-19 pandemic has been described (8) and is innate to health care. ~~However, but~~ fear among DCS staff due to workloads ~~or and~~ sudden changes in routine following mitigation strategies has not been described during this pandemic.

Line 247-248: Try to avoid using 'and' multiple times in a sentence

Concerns about contracting or spreading COVID-19 among DCS staff at work, even though we found the highest risk of infection when living in a household with an infected individual, may be related to suboptimal communication and education from the authorities providing the mitigation strategies ~~which and~~ could be a focus point in future strategies.

Line 254: use a comma instead of a ;

The possibility of misclassification bias as participants read the POCT themselves is one of the study's limitations, ~~;~~ this could result in false negative or false positive test results.

Line 254: After the comma (see above), substitute 'which' instead of 'this'

The possibility of misclassification bias as participants read the POCT themselves is one of the study's limitations, ~~which this~~ could result in false negative or false positive test results.

Line 272-274: Try to avoid using 'and' multiple times in a sentence

The contrast between perceived fear (being infected at work) and measured risk (being infected in the household) represents an important issue for risk communication **as well as and** pandemic planning.

Line 297 - 298: Try to avoid using 'and' multiple times in a sentence

All authors took part in conceptualization, interpretation, **and** discussion of results, agree to be accountable for all aspects of the work and approved the final version of the manuscript.

Reviewer #3:

- 1. Though not statistically significant ($p=0.06$), there is a tendency that more participants wore facemasks in seronegative group than seropositive (Figure 3). I would like to know how the authors would consider this tendency and usefulness of wearing masks.**

Response to Reviewer, comment 1:

Thank you for this relevant question.

Employees in 'day-care facilities and pre-schools' (DCS) could use a face shield at work to protect themselves or others, but the use of facemasks has not been recommended for DCS due to the children's ability to see and understand the staff's facial expressions.

Facemasks were, among other things) recommended for use in public transportation and grocery stores at the time of the study. People who use facemasks more frequently in society may also protect themselves more frequently at work, and thus are not only seronegative due to the use of face masks at work, but also in general, which reduces the risk of becoming infected.

Prior to revision:

In our study, less than 25% of participants wore face masks at work, while even fewer participants wore face shields. As previously stated, DCS employers were not required to wear facemasks at work, but they could use a face shield if they wished, as small children, in particular, will need to see and read facial expressions and body language when in contact with adults.

Revision (Discussion, page 13, l. 229):

In our study, less than 25% of participants wore face masks at work, while even fewer participants wore face shields, **with a tendency that more seronegative than seropositive participants wore facemasks**. As previously stated, DCS employers were not required to wear facemasks at work, but they could use a face shield if they wished, as small children, in particular, will need to see and read facial expressions and body language when in contact with adults. **Facemasks were recommended for use in public transportation and grocery stores at the time of the study. People who use facemasks more frequently in society may also protect themselves more frequently at work, and thus are not only seronegative due to the use of face masks at work, but also in general, which reduces the risk of becoming infected.**

Added to the manuscript (Discussion, page 13, l. 224):

Despite a high level of adherence to national recommendations (supplementary figure 1), no discernible difference was found between individual protective health measures and serostatus, **as seen in earlier studies (ref).**

Ref: Fogh K, Strange JE, Scharff B, Eriksen ARR, Hasselbalch RB, Bundgaard H, et al. Testing Denmark: a Danish Nationwide Surveillance Study of COVID-19. Microbiol Spectr. 2021;9(3):e0133021.

Ref: Bundgaard H, Bundgaard JS, Raaschou-Pedersen DET, von Buchwald C, Todsén T, Norsk JB, et al. Effectiveness of Adding a Mask Recommendation to Other Public Health Measures to Prevent SARS-CoV-2 Infection in Danish Mask Wearers : A Randomized Controlled Trial. Ann Intern Med. 2021;174(3):335-43.

- 2. Relevant to comment #1, I would like to see the Figure comparing seropositive/negative ratio among those with/without following each preventive intervention.**

Response to Reviewer, comment 2:

This is a relevant suggestion, thank you. The questionnaire was designed so that participants could indicate whether they had changed their behavior due to the risk of infection with COVID-19. They could tick 1 or more of the answers below: Cough/sneeze in sleeve, frequent handwash, use face shield, use face mask or none of the above.

See the two figures below which present the information presented in Figure 3 in the manuscript from different angles. Both figures can be attached to the manuscript as supplementary figures. However, in our opinion, they add no new information not already shown in Figure 3.

The first figure below depicts the seronegative and seropositive proportion of individuals who have undertaken preventive measures - or have taken none of the mentioned. The second one shows the amount of seropositive and seronegative individuals who have undertaken preventive measures (or none).

3. **Methods; As in Reference #18&22, sensitivity and specificity of both assays should be described. It will help readers interpreting the results.**

Response to Reviewer, comment 3:

Thank you for this suggestion. We have added the sensitivity and specificity for both assays to the manuscript.

Revision (page 7, l. 94):

POCT sensitivity and specificity were reported to be 96.9% (95% CI 96.7-98.5%) and 99.4% (95% CI 97.8-99.8%) by the manufacturer, respectively (ref). A comparative study (cases = 30 people, controls = 10) revealed a slightly lower sensitivity of 90.0% and a 100% specificity (ref). A study of 129 non-hospitalized versus 31 hospitalized SARS-CoV-2 patients revealed that the Wantai assay had a sensitivity of 96.7 (95% CI 92.4-98.6) and a specificity of 99.5 (95% CI 98.7-99.8) (ref).

Ref: CTK Biotech inc. OnSite COVID-19 IgG/IgM rapid test. 2020. <https://cdn2.hubspot.net/hubfs/4418101/PI-R0180C%20Rev%20B2.1.pdf>.

Ref: Lassaunière R, Frische A, Sværke Jørgensen C. Evaluation of nine commercial SARS-CoV-2 immunoassays. medRxiv, preprint version. 2020.

Ref: Harritshøj LH, Gybel-Brask M, Afzal S, Kamstrup PR, Jørgensen CS, Thomsen MK, et al. Comparison of 16 serological SARS-CoV-2 immunoassays in 16 clinical laboratories. *J Clin Microbiol.* 2021;59(5):e02596-20. doi: 10.1128/JCM.02596-20.

4. **Table 1; the calculation of the proportion is wrong (for example; Previously positive PCR test among Seropositive = 1066/1857 = 57.4% but 61.1% in Table 1). The table needs revision.**

Response to Reviewer, comment 4:

Thank you very much for pointing this important detail out.

We did not state the number of "missing values" in table 1 because there were varying numbers of missing values for the different variables.

We have updated the table title in the hopes that the results are clearer, and our method is more transparent. We have also made a small addition to the methods section that describes how we handled missing data in greater detail.

Added to the manuscript (Method section, Statistical analysis, page 9, l. 126):

Individuals were also excluded from analysis of questionnaire questions that they had not answered.

Prior to revision, Title, Table 1:

Table 1: Baseline characteristics of the study cohort on age, sex, BMI, smoking, alcohol use, education level and place of work stratified by seropositivity among 20,267 unvaccinated participants

Revision, Title, Table 1:

Table 1: Baseline characteristics of the study cohort on age, sex, BMI, smoking, alcohol use, education level and place of work stratified by seropositivity among 20,267 unvaccinated participants. **Missing values are excluded from analysis.**

December 6, 2022

Dr. Kamille Fogh
Herlev and Gentofte Hospital, University of Copenhagen
Dept. of Cardiology & Dept. of Emergency Medicine
Herlev
Denmark

Re: Spectrum04174-22R1 (A cross-sectional study of SARS-CoV-2 antibodies and risk factors for seropositivity in staff in day-care facilities and pre-schools in Denmark)

Dear Dr. Kamille Fogh:

Thank you for submitting your revised manuscript and thoroughly addressing reviewer concerns. Your manuscript has been accepted, and I am forwarding it to the ASM Journals Department for publication. You will be notified when your proofs are ready to be viewed.

Sincerely,

Holly Ramage
Editor, Microbiology Spectrum
